# Detection as Regression: Certified Object Detection by Median Smoothing

**Ping-yeh Chiang**
University of Maryland
pchiang@cs.umd.edu

**Michael J. Curry**
University of Maryland
curry@cs.umd.edu

**Ahmed Abdelkader**
University of Maryland
akader@cs.umd.edu

**Aounon Kumar**
University of Maryland
aounon@cs.umd.edu

**John Dickerson**
University of Maryland
john@cs.umd.edu

**Tom Goldstein**
University of Maryland
tomg@cs.umd.edu

## Abstract

Despite the vulnerability of object detectors to adversarial attacks, very few defenses are known to date. While adversarial training can improve the empirical robustness of image classifiers, a direct extension to object detection is very expensive. This work is motivated by recent progress on certified classification by randomized smoothing. We start by presenting a reduction from object detection to a regression problem. Then, to enable certified regression, where standard mean smoothing fails, we propose median smoothing, which is of independent interest. We obtain the first model-agnostic, training-free, and certified defense for object detection against $\ell_2$-bounded attacks.

## 1 Introduction

Adversarial examples are seemingly innocuous neural network inputs that have been deliberately modified to produce unexpected or malicious outputs. Early work on adversarial examples was highly focused on image classifiers, which assign a single label to an entire image [13, 6, 25, 8]. A large literature has rapidly emerged on defenses against classifier attacks, which includes both heuristic defenses [23] and certified methods with theoretical guarantees of robustness [9, 36, 2, 24, 26]. However, most realistic vision systems crucially rely on *object detectors*, rather than *image classifiers*, to identify and localize multiple objects within an image [29, 28, 21].

Over time, attacks on object detection have become more sophisticated, as has been successfully demonstrated both digitally and in the physical world using a range of perturbation techniques, as well as attacks that break both the object localization and classification parts of the detection pipeline [22, 32, 37, 15, 19]. As of this writing, we are only aware of one recent paper on the adversarial robustness of object detectors [39]. This lack of defenses is likely because (i) the complexity of the multi-stage detection pipeline [29] is difficult to analyze, and (ii) detectors are far more expensive to train than classifiers. Furthermore, (iii) object detectors output bounding-box coordinates, and are thus *regression* networks to which many standard defenses for *classifier* networks cannot be readily applied.

In this paper, we present, to the best of our knowledge, the first certified defense against adversarial attacks on object detectors. To avoid the difficulties discussed above, we treat the complex detection pipeline as a black box without requiring specialized re-training schemes. In particular, we present a reduction from object detection to a single regression problem which envelopes the proposal, classification, and non-maximum suppression stages of the detection pipeline. Then, we endow this regression with certified robustness using the Gaussian smoothing approach [4], originally proposed

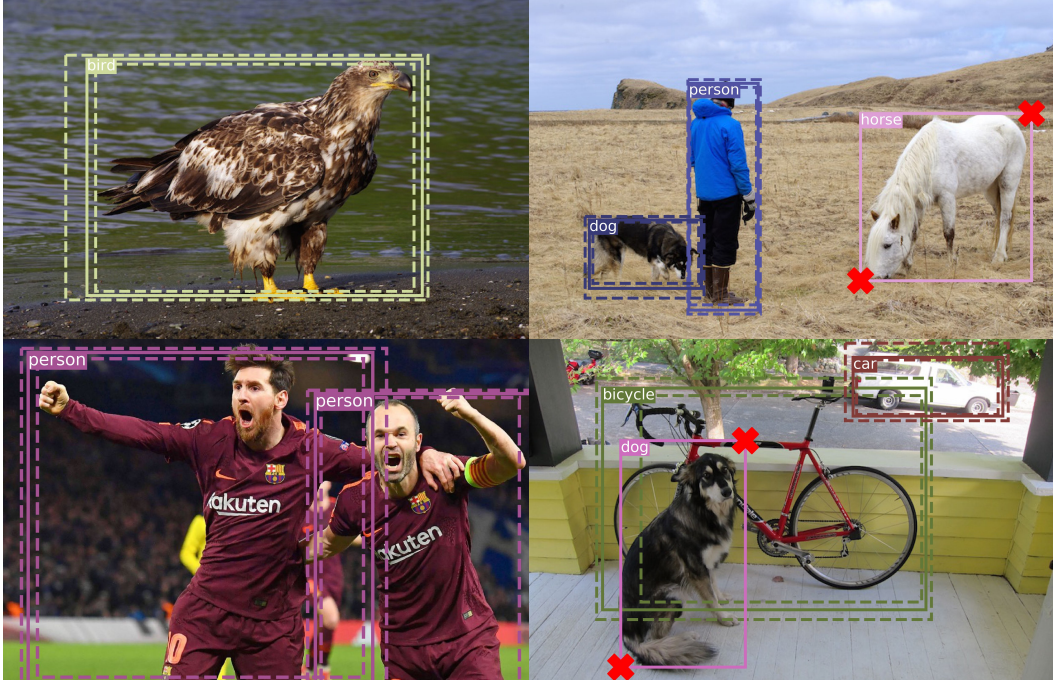

Figure 1: Samples of object detection certificates using the proposed method. Dotted lines represent the farthest a bounding box could move under an adversarial perturbation $\delta$ of bounded $\ell_2$-norm. If the predicted bounding box can be made to disappear, or if the label can be made to change, after a perturbation with $\|\delta\|_2 < 0.36$, then we annotate the bounding box with a red X.

for the defense of classifiers. To this end, we develop a new variant of smoothing specifically for regression based on the *medians* of the smoothed predictions, rather than their mean values. The proposed *median smoothing* approach enjoys a number of useful properties, and we expect it to find further applications in certified robustness. Finally, we implement our method to obtain a *certifiably-robust* wrapper of the YOLOv3 [28], Mask RCNN [10], and Faster RCNN [29] object detectors. We use the MS-COCO dataset [20] to test the resulting detector, obtaining the first detector to achieve non-trivial $\ell_2$-norm certified average precision on large scale image dataset.

## 2   Background

We briefly review attacks on object detectors, and the certified classification methods we build upon.

**Attacks and defense on object detection and semantic segmentation.** Attacks exist that interfere with different components of the detection pipeline. Dense Adversary Generation (DAG) is an early attack that interferes with the classifier stage of detection [38], and was later extended to videos [35]. In contrast, region proposal networks can be manipulated by decreasing the confidence of proposals [17]. The DPatch attack causes misclassification by placing a patch that does not overlap with the objects of interest [22], while the attack in [18] contaminated images with "imperceptible" patches.

The attacks described above are all digital. Early physical-world attacks on detectors fooled stop sign detectors by modifying the entire stop sign [5, 3]. While it was shown that detectors are much more robust to attacks than classifiers [29], later works successfully broke detectors using patch attacks that do not require whole-object modification. This includes printed adversarial patches that deceive person detectors [34], and adversarial clothing that makes its wearer invisible to a range of detection systems [37].

Despite a plethora of attacks, we are only aware of a single paper studying adversarial defenses for object detectors [39]. In [39], they adversarially train the object detector base on both the classification and localization loss. While empirically effective, such an approach could fail against

stronger, more sophisticated attackers whereas as our approach can guarantee robustness against all possible attackers within the threat model.

In this paper, we present a *certified* defense for object detectors that is robust regardless of the method used to craft the attack. Our work is motivated by recent progress on certified image classification by the randomized smoothing approach, as we review next.

**Certified defenses for image classification.** Several methods of obtaining robustness certificates for classification problems have been proposed [24, 26, 11, 27, 7, 40, 33]. In addition, [14, 16, 30, 31, 36] proposed methods to both defend the model while enabling better certificates. For our purposes, we focus on randomized smoothing defenses [4, 14]. These first convert a base classifier to a smoothed classifier by labeling many images sampled from a Gaussian ball around the input, and taking a majority vote [4]. The effect of image perturbations on this smoothed classifier can be bounded using either the Neyman-Pearson Lemma [4] or properties of the Weierstrass transform [30].

Since [14] proposed randomized smoothing for robustness, it has been improved by tightening the certificate [4], improving the training process [30], adapting it to new threat models [16], or incorporating confidence level [12] – all in the context of classification tasks. In particular, the voting scheme in [4, 12] requires a global bound on the output of the base classifier, which naturally holds for binary classifiers with a 0/1 output.

We will see later that existing certificates based on randomized smoothing become weaker when applied to regression problems over a large range of output values. Towards the intended application of robust detection, we propose a new certificate for regression problems based on the median of the smoothed predictions, rather than their mean values, which is of independent interest.

## 3 Detection Certificates

In this section, we introduce the proposed certificate in the context of modern detection pipelines. We begin by describing the black box interface that allows us to certify the outputs of detector networks without the need for re-training. Then, we motivate and define the proposed certificate at a high level, where the next two sections fill in the details.

**Detection interface.** Typical object detectors take an input image and output a variable-length list of bounding boxes and associated class labels $\{(b_1, \ell_1), (b_2, \ell_2), \dots \}$. State-of-the-art detectors, such as Faster-RCNN [29], YOLO [28], and RetinaNet [21] usually have many output heads, each of which is responsible for a single bounding box. As the number of output heads is usually much larger than the number of objects, redundant boxes need to be filtered out, e.g., by *Non-Maximum Suppression* (NMS). First, NMS discards any box with an *objectness score* below a threshold $\tau$. Then, the box $b^*$ with the highest score is taken out as output, and any remaining boxes overlapping $b^*$ significantly are filtered out. This process is repeated until there are no more boxes left.

**Certifying detector outputs.** Adversarial attacks on object detectors attempt to distort the location and appearance of objects. We certify detector outputs by bounding the positions and sizes of the detected objects. In addition, we ensure that the associated class labels stay the same.

**Representation.** We represent a bounding box $b$ using the coordinates of its corners $(x_1, y_1, x_2, y_2)$, with $x_1 \leq x_2$ and $y_1 \leq y_2$, along with the associated class label $\ell$ and objectness score. Then, to measure the overlap between two boxes $b_1$ and $b_2$, we use the *Intersection over Union* defined as $\text{IoU}(b_1, b_2) = \text{Area}(b_1 \cap b_2)/\text{Area}(b_1 \cup b_2)$.

**Bounding-box certificate.** Given a predicted bounding box $b$, we aim to certify its size and location. Assume for now that we obtained certified lower and upper bound for each coordinate of $b$ as $(\underline{x}_1, \underline{y}_1, \underline{x}_2, \underline{y}_2)$ and $(\overline{x}_1, \overline{y}_1, \overline{x}_2, \overline{y}_2)$. We say that a box is "certifiably correct" if the IoU between the ground truth bounding box and the worst-case bounding box, with coordinates respecting the certified bounds, is above a certain threshold. The worst-case bounding box is the box with coordinates satisfying the certified upper and lower bounds which realizes the lowest IoU with the ground-truth box. If $\underline{x}_2 \leq \overline{x}_1$ or $\underline{y}_2 \leq \overline{y}_1$, then the worst-case IoU is zero. Otherwise, the worst-case bounding box can always be found in the set $\{\underline{x}_1, \overline{x}_1\} \times \{\underline{y}_1, \overline{y}_1\} \times \{\underline{x}_2, \overline{x}_2\} \times \{\underline{y}_2, \overline{y}_2\}$. Hence, we simply

enumerate all 16 boxes, and take the smallest IoU. As long as the worst-case IoU is larger than the threshold $\tau$, then the box $b$ is considered certifiably correct.

**Label certificates.** We treat the label $\ell \in \mathbb{N}$ as an additional coordinate. Again, assuming we obtained certified lower and upper bounds as $\overline{\ell}$ and $\underline{\ell}$, then $\ell$ is only considered certified when $\overline{\ell} = \underline{\ell}$.

In the next section, we describe the smoothing approach we use to obtain the required certified bounds on each coordinate.

## 4 Median Smoothing for Regression

A number of strategies have been proposed for certifying classifiers, many based on Gaussian means. We will see below that the bounds provided become fairly weak for regression problems. For this reason, we propose smoothing based on Gaussian *medians*, which provide considerably stronger bounds for regression networks such as object detectors.

**Mean smoothing.** Given a base function $f : \mathbb{R}^d \to \mathbb{R}$, its Gaussian smoothed analog is [4, 14, 30]

$$g(x) = \mathbb{E}[f(x + G)], \quad \text{where } G \sim N(0, \sigma^2 I). \tag{1}$$

In the context of classifier networks, where output heads take the form $f : \mathbb{R}^d \to [0, 1]$, a certificate can be obtained by bounding the gap between the highest and second-highest class probabilities [4]. However, we aim to apply the smoothing technique to a general regression problem, such as bounding box regression, with $f : \mathbb{R}^d \to [l, u]$. In that case, the bound on $g$ can be rather loose, which follows by invoking Lemma 2 from [30] (see their appendix) with the normalized function $\frac{f(x)-l}{u-l}$ as stated below; see Appendix A for further discussion. Throughout, we use $\Phi$ to denote the cumulative distribution function (CDF) of the standard Guassian distribution.

**Corollary 1.** *[30] For any $f : \mathbb{R}^d \to [l, u]$, the map $\eta(x) = \sigma \cdot \Phi^{-1}\left(\frac{g(x)-l}{u-l}\right)$ is 1-Lipschitz, implying*

$$l + (u - l) \cdot \Phi\left(\frac{\eta(x) - \|\delta\|_2}{\sigma}\right) \leq g(x + \delta) \leq l + (u - l) \cdot \Phi\left(\frac{\eta(x) + \|\delta\|_2}{\sigma}\right) \tag{2}$$

**Median smoothing.** The issue with Gaussian smoothing is that the mean values it computes can be highly skewed by extreme values of the base function. Hence, the resulting bounds are rather loose when applied to functions with large variations in their outputs. This is not a problem for classifiers (which output values between 0 and 1), but is highly problematic for general regression problems.

To obtain tighter certificates, we utilize the percentiles of the output random variable instead of its mean. In particular, as the *median* is almost unaffected by outliers, a global bound on the base function is no longer required. Formally, we propose the following formulation.

**Definition 1.** *Given $f : \mathbb{R}^d \to \mathbb{R}$ and $G \sim N(0, \sigma^2 I)$, we define the* percentile smoothing *of $f$ as*

$$\underline{h}_p(x) = \sup\{y \in \mathbb{R} \mid \mathbb{P}[f(x + G) \leq y] \leq p\} \tag{3}$$

$$\overline{h}_p(x) = \inf\{y \in \mathbb{R} \mid \mathbb{P}[f(x + G) \leq y] \geq p\} \tag{4}$$

While the two forms $\underline{h}_p$ and $\overline{h}_p$ are equivalent for continuous distributions, the distinction is needed to handle edge cases with discrete distributions. In the remainder of the paper, we will use $h_p$ to denote the percentile-smoothed function when either definition can be applied. While $h_p$ may not admit a closed form, we can approximate it by Monte Carlo sampling [4], as we explain in Section 5.

A useful property of percentile smoothing is that it always outputs a realizable output of the base function $f$. This can be useful when $f$ produces discrete values or labels (as is the case for classifiers), or bounding boxes. In contrast, mean smoothing is a weighted average of the outputs, and it is more susceptible to outliers. For example, when two distinct bounding boxes are predicted, we select one or the other rather than their average.

**Regression certificates.** To certify a percentile-smoothed function $h_p$ for input $x$ under adversarial perturbations of bounded $\ell_2$-norm, we evaluate the function at $x$ with two appropriate percentiles

$p$ and $\overline{p}$. The basic idea is to first bound the probability that the output of the base function $f$ will fall below a particular threshold $\Lambda$. A key observation is that this is equivalent to bounding a mean-smoothed indicator function $\mathbb{E}(\mathbf{1}_{f(x+G)<\Lambda})$, which satisfies the assumption of Corollary 1. We can then use this bound to further bound the change in the percentiles output by $h_p$; see Appendix B for the full proof.

**Lemma 1.** *A percentile-smoothed function $h_p$ with adversarial perturbation $\delta$ can be bounded as*

$$\underline{h_p}(x) \leq h_p(x+\delta) \leq \overline{h_{\overline{p}}}(x) \quad \forall \|\delta\|_2 < \epsilon, \tag{5}$$

*where $\underline{p} := \Phi\left(\Phi^{-1}(p) - \frac{\epsilon}{\sigma}\right)$ and $\overline{p} := \Phi\left(\Phi^{-1}(p) + \frac{\epsilon}{\sigma}\right)$, with $\Phi$ being the standard Gaussian CDF.*

The immediate benefit of $h_p$ is that the tightness of the bound now depends on the concentration of $f$ around the $p$-th percentile of $f(x+G)$, per the local gap between the percentiles $\underline{h_p}(x)$ and $\overline{h_{\overline{p}}}(x)$. In contrast, the bound obtained by $g$ depends on the position of $\mathbb{E}[f(x+G)]$ relative to the global bounds per the extreme values $l$ and $u$.

In addition, percentile smoothing can be applied without specifying the bounds $l$ and $u$, which may be unknown a priori. Even when one or both of the bounds $l$ and $u$ is infinite, percentile smoothing provides a non-vacuous certificate where mean smoothing fails. For example, take $f$ so that $f(x+G) \sim N(0,1)$. In this case, it is easy to see that the bounds on $g(x+\delta)$ per Corollary 1 are vacuous, while $h_{50\%}(x+\delta)$ can be bounded between $\pm\|\delta\|_2/\sigma$ by Lemma 1.

Now that we have a mechanism for bounding the individual coordinates output by an object detector, we proceed to describe the specifics of our approach and its implementation.

## 5   Implementing Detection Certificates

In order to certify the predictions output by the object detector, we aggregate multiple predictions made under randomly perturbed inputs using the median smoothing approach presented in Section 4. Roughly speaking, the more the aggregated predictions agree, the stronger the certificate.

Recall that each prediction consists of four coordinates $(x_1, y_1, x_2, y_2)$ representing a bounding box, with the associated label $\ell \in \mathbb{N}$ treated as the fifth coordinate. As outlined in Section 3, the proposed certificate requires lower and upper bounds for each coordinate of each prediction.

Our certification strategy is based on treating each coordinate $c$ as the output of a dedicated function $f_c$. Then, we apply median smoothing to certify each coordinate independently. The challenge to implementing this strategy is to consolidate multiple predictions, each with five coordinates, as a single vector so that certified regression can be applied. The main complication is that object detectors typically produce a variable-length list of predictions in no particular order. In contrast, our regression model requires a single vector with a consistent assignment of indices to prediction coordinates.

**Certification pipeline.**   First, we compose a base detector $b$ into a sequence $f = r \circ b \circ d$, where $r$ encodes the predictions produced by $b$ into one or more vectors, and $d$ is a denoiser to improve concentration. Then, we work with the median smoothing of $f$, see Definition 1, and use Monte Carlo sampling to approximate $h_p(x)$ along with lower and upper bounds. See Figure 2 for the overall workflow and Algorithm 1 for the pseudocode.

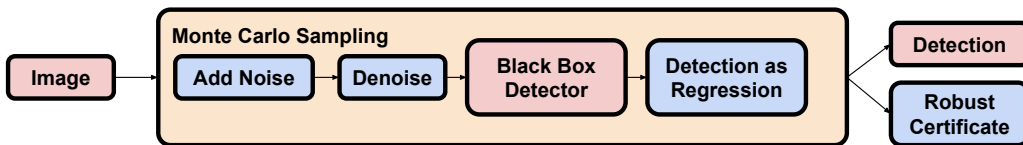

Figure 2: Converting a base detector to a certified robust detector.

**Denoising.**   The denoiser $d$ is applied to the input of the base detector $b$. The idea is that, since the Gaussian smoothing certificate can be applied to any pipeline, we might as well apply it to one that begins with a denoiser that removes most of the Gaussian noise. This makes $f(x+G)$ more concentrated [31], resulting in a stronger certificate without re-training on noisy data.

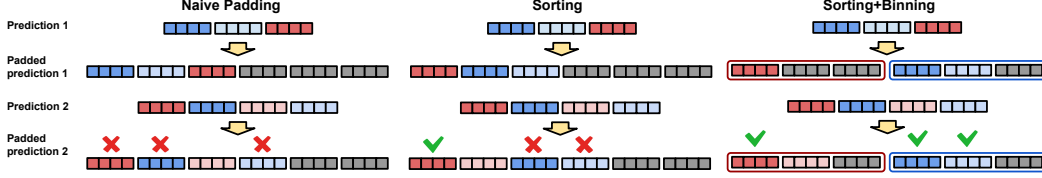

Figure 3: Even when the bounding boxes predicted for one image appear among the predictions for another noisy sample, the outputs will appear completely different if the boxes are not ordered in the same way. In the first column, we convert the detected output into a fixed length vector by directly padding the output, and none of the boxes are aligned. In the second column, we sort the boxes by location first before padding, allowing us to align one box out of three. In the third column, we place the boxes into corresponding bins before sorting, and this approach aligns all three boxes as desired.

**Encoding detectors as regressors.** Given $n$ detections of potential objects, the corresponding predictions can be represented as $u \in \mathbb{R}^{n \times 5}$, where $n$ may vary across random perturbations. We aim to convert $u$ into a suitable vector $v = r(u)$. A naive approach to implement the encoding $r$ is to simply copy $u$ into $v$ before possibly padding $v$ with sentinel entries up to the desired length. However, even if the detector produces the same predictions under different noises, their ordering may be different leading to inconsistencies that weaken the certificate.

To improve the consistency of the encoded regression vectors, we propose two operations. The first operation is based on *sorting* the predictions in $u$ either by their objectness scores or by the centers of their bounding boxes.[1] The second operation is based on partitioning $u$ into independent *bins* based on the labels or the locations[2] of the predictions, with each bin $b_i$ encoded separately as a vector $v_i$. Detecting a different number of objects in some bin does not affect the other bins; see Figure 3. For example, when binning by label, a dog which is only detected under some perturbations would not impact the certificates produced for any cars; see Figure 1. As the random perturbations may produce vectors of varying lengths, we implicitly assume that all vectors are padded with sentinel values, taken as $\infty$, such that every coordinate has a corresponding realization in all outputs.

---

**Algorithm 1** Prediction and Certified Detection

---

**function** DETECT($f, x, \sigma, n$)
    $\hat{x} \leftarrow$ AddGaussianNoise($x, \sigma, n$)             $\triangleright$ $\hat{x}$ is $n$ times as large as $x$
    $\hat{y} \leftarrow$ toRegression($f(\hat{x})$)      $\triangleright$ Convert detection output into regression vectors
    $\hat{y} \leftarrow$ Sort($\hat{y}$)      $\triangleright$ Sort each coordinates along the batch dimension
    $y_{median} \leftarrow \hat{y}_{\lfloor 0.5n \rfloor}$             $\triangleright$ Take the median
    $bbox, \ell \leftarrow$ toDetection($y_{median}$)
    **return** $bbox, \ell$
**function** CERTIFYDETECT($f, x, \sigma, n, \epsilon, c$)
    $q_u, q_l \leftarrow$ GetEmpiricalPerc($n, \epsilon, c$)
    $\hat{x} \leftarrow$ AddGaussianNoise($x, \sigma, n$)             $\triangleright$ $\hat{x}$ is $n$ times as large as $x$
    $\hat{y} \leftarrow$ toRegression($f(\hat{x})$)
    $\hat{y} \leftarrow$ Sort($\hat{y}$)      $\triangleright$ Sort each coordinates along the batch dimension
    $y_{median} \leftarrow \hat{y}_{\lfloor 0.5n \rfloor}$             $\triangleright$ Take the median
    $y_u, y_l \leftarrow \hat{y}_{q_u}, \hat{y}_{q_l}$             $\triangleright$ Take the $q$th order statistics
    $bbox, \ell, \overline{bbox}, \overline{\ell}, \underline{bbox}, \underline{\ell} \leftarrow$ toDetection($y_{median}, y_u, y_l$)
    **return** $bbox, \ell, \overline{bbox}, \overline{\ell}, \underline{bbox}, \underline{\ell}$

---

**Evaluation of $h_p$ (GetEmpiricalPerc).** In practice, we resort to using Monte Carlo sampling to approximate the upper bound of $\overline{h}_{\overline{p}}(x)$ and lower bound of $\underline{h}_{\underline{p}}(x)$ similar to [4]. Given $n$ draws

$\{G_1, G_2, \ldots, G_n\}$, we evaluate $X_i = f(x + G_i) \in \mathbb{R}$. Then, we find the corresponding order statistics $0 = K_0 \leq K_1 \leq K_2 \cdots \leq K_n \leq K_{n+1} = \infty$. We want to find the empirical order statistic $K_{q^u}$ and $K_{q^l}$, such that $P(K_{q^u} \geq \overline{h}_{\overline{p}}) \geq \alpha$ and $P(K_{q^l} \leq \underline{h}_{\underline{p}}) \geq \alpha$. Specifically, given an order statistic $K_{q^u}$, we can evaluate $P(K_{q^u} \geq \overline{h}_{\overline{p}})$ explicitly using the binomial formula below.

$$P(\overline{h}_{\overline{p}} \leq K_{q^u}) = \sum_{i=1}^{i=q^u} P(K_{i-1} < \overline{h}_{\overline{p}} \leq K_i) = \sum_{i=1}^{j=q^u} \binom{n}{i} (\overline{p})^i (1 - \overline{p})^{n-i} \tag{6}$$

A similar formula can be derived for $P(\underline{h}_{\underline{p}} \geq K_{q^l})$. We can then use binary search to find the smallest $q^u$ and largest $q^l$ such that $P(\overline{h}_{\overline{p}} \leq K_{q^u}) \geq \alpha$ and $P(\underline{h}_{\underline{p}} \geq K_{q^l}) \geq \alpha$ are satisfied.

## 6 Experiments

We mainly use YOLOv3 [28], pretrained on the MS-COCO dataset [20], as our black-box detector where IoU thresholds for NMS is set to 0.4. The evaluation is done on all 5000 images from the test set, with adversarial perturbations $\|\delta\|_2 < \epsilon = 0.36$. We set the IoU threshold for certification $\tau = 0.5$, as it is a common setting for evaluation on the MS-COCO dataset [20]. To perform the smoothing, we inject Gaussian noise with standard deviation $\sigma = 0.25$, and we use 2000 noise samples for each image. The estimated upper and lower bound for each coordinate are selected such that they bound the true $\overline{h}_{\overline{p}}(x)$ and $\underline{h}_{\underline{p}}(x)$ with confidence $\alpha = 99.999\%$. For denoising, we use the DNCNN denoiser [41] pretrained by [31] (with $\sigma = 0.25$).

We find all three operations – sorting, binning, and denoising – are helpful in mitigating the impact of smoothing on the clean performance as well as increasing the certified performance. All of these methods complement each other for the most part, and we are able to achieve the most robust and accurate smoothed object detector by using all three methods. Specifically, we find that sorting by box location works best, and that prepending a denoiser is indeed very important for both the clean and certified performance.

**Robustness metrics and performance evaluation.** We evaluate the smoothed models based on two metrics: AP and certified AP. AP is indicative of the smoothed model's performance in absence of an adversary, and ideally, we would like to avoid drops in AP when converting the base detector to a smoothed one. On the other hand, certified AP tells us the guaranteed lower bound on the AP when faced with the specified adversary.

More specifically, *certified precision and recall* are calculated as follows. To get the number of certifiably correct boxes, we count the number of detections whose worst-case IoU with the corresponding ground-truth box exceeds the threshold $\tau$. To calculate the certified precision, we also need to know the maximum possible number of detections under any perturbed input. We upper bound this number by counting all certifiably non-empty entries across all regression vectors.

$$\text{Certified Recall} = \frac{\text{\# certifiably correct detections}}{\text{\# ground-truth detections}} \tag{7}$$

$$\text{Certified Precision} = \frac{\text{\# certifiably correct detections}}{\text{max \# predicted detections}} \tag{8}$$

Due to the binning and sorting processes associated with smoothing, it is difficult to calculate the exact precision/recall curve at all objectness thresholds. We instead evaluate certified precision and recall at 5 different objectness thresholds $\{0.1, 0.2, 0.4, 0.6, 0.8\}$, and use the area under the steps to lower-bound the true certified AP.

**Sorting and binning methods.** We find that sorting bounding boxes by location consistently achieves better clean AP and certified AP than sorting them by objectness score, as shown in Table 1. The only exception is when location binning is used together with location sorting where the clean AP improves, but certified AP decreases. In the best case, switching to location sorting improves AP by $4.52\%$ and certified AP by $0.64\%$.

| Binning Method | Sorting Method | AP @ 50 | Certified AP @ 50 |
|---|---|---|---|
| Blackbox detector | | 48.66% | - |
| None | Objectness | 17.60% | 1.24% |
| | Location | 21.51% | 1.24% |
| Label | Objectness | 25.27% | 2.67% |
| | Location | 29.75% | 3.32% |
| Location | Objectness | 27.48% | 3.23% |
| | Location | 28.90% | 2.67% |
| Location+Label | Objectness | 30.32% | 3.97% |
| | Location | 32.04% | 4.18% |

Table 1: Clean and certified AP using various sorting and binning methods with YOLOv3 as base detector. Detailed precision/recall statistics can be found in Appendix D.

Both label binning and location binning also consistently boost the clean AP and certified AP. Compared to no binning, label binning alone could increase the AP by 7-8% and the certified AP by 1-2%, as shown in Table 1. Furthermore, the two binning methods complement each other, and we are able to achieve the best AP and certified AP by combining both location binning and label binning.

**Denoising.**    Because we do not retrain the detector under noise, denoising the image first is extremely important in achieving good clean AP and certified AP. In Table 2, we present performance with and without the denoiser – given the results in Table 1 as summarized above, we restrict experiments to the location sorting method. Without the denoiser, the clean AP drops by an average of 20.07% and the certified AP drops by an average of 2.55%.

| Binning Method | Denoise | AP @ 50 | Certified AP @ 50 |
|---|---|---|---|
| None | Yes | 21.51% | 1.24% |
| | No | 5.85% | 0.14% |
| Label | Yes | 29.75% | 3.32% |
| | No | 8.58% | 0.32% |
| Location | Yes | 28.90% | 2.67% |
| | No | 8.03% | 0.32% |
| Location+Label | Yes | 32.04% | 4.18% |
| | No | 9.49% | 0.44% |

Table 2: Denoising significantly improves the clean AP and certified AP with YOLOv3 as base detector. Sorting method: location. Detailed precision/recall statistics can be found in Appendix D.

**Architecture.**    Since our proposed approach treats the base detector as a black box, we also experimented with Mask-RCNN and Faster-RCNN as the base detector. We use the best combination of settings from Table 1 for all three architectures. Surprisingly, even though the base Mask-RCNN and Faster-RCNN perform better compared to YOLOv3, after smoothing, they both perform consistently worse. Certified AP is decreased by almost 2/3 after switching to the alternative architecture.

**Tightness of certification.**    To understand the tightness of our certification, we implemented the DAG attack [38] against our most robust smoothed detector. We take 20 PGD steps and draw 5 random samples to estimate the gradient of the smoothed model. Surprisingly, the smoothed model is quite robust within the desired radius. The DAG attack was only able to decrease recall by 1.1%. This illustrates that the bound we obtained is likely quite loose with respect to the true robustness of the smoothed object detector.

| Architecture | Base Detector<br>AP @ 50 | Smoothed Detector | |
|---|---|---|---|
| | | AP @ 50 | Certified AP @ 50 |
| YOLOv3 | 48.66% | 31.93% | 4.21% |
| Mask RCNN | 51.28% | 30.53% | 1.67% |
| Faster RCNN | 50.47% | 29.89% | 1.54% |

Table 3: Robustness comparison using different base detectors.

**Inference speed.** We note that the Monte Carlo sampling process is inherently costly. In our experiments, we used 2000 samples to approximate the smooth model for each image, which makes our evaluation 2000 times as expensive. This is a common problem in the randomized smoothing approach [4], and reducing the sample complexity is still an active area of research [1].

While the certified AP are still far below the requirements of real-world applications, the proposed smoothing approach is able to achieve non-trivial certified AP *without* any re-training of the base detector. We think this is a promising direction for eventually obtaining practical verifiably robust object detectors.

# 7 Conclusion and Future Work

We propose a new type of randomized smoothing using a percentile instead of an expectation, such that certificates can be easily generated for regression-type problems. We then apply it to obtain the first certified defense for object detectors. Some potential future work includes decreasing the sample complexity of our randomized certificate, extending our robust detector to defend against other threat models like patch attacks leveraging the method from [16], or using a learning approach to find a better reduction from detection to regression compared to our binning/sorting approach. We note that the machinery we have developed for certifying regression problems is largely application agnostic, and we hope it can find use in certifying a range of other regression tasks.

# Broader Impact

Neural networks are very powerful tools, and society will benefit greatly if they can be used in a broader range of safety critical applications. In particular, object detectors can be used in many systems that must visually perceive and interact with the real world – perhaps most strikingly, in autonomous vehicles. Ensuring the safety and predictability of neural networks is critical in enabling the application of neural networks in these areas, and certificates associated with safety with respect to a particular model are very useful in providing assurance that the neural network cannot be exploited by malicious actors. At the same time, there is real concern about the privacy impact of widespread deployment of modern computer vision systems, including systems like object detectors and face recognition systems that produce bounding boxes. If individuals wish to use physical or digital adversarial examples to protect their privacy, the techniques we present might make it more difficult for them to do so, although it is not actually clear that adversarial examples will ultimately prove effective or useful for protecting privacy. In any case, we are not aware of any real-world adversarial attacks being performed "in the wild", for good or ill. We believe the concrete positive impact on safety is probably greater than a hypothetical negative impact on privacy.

# Acknowledgments

Goldstein and Chiang were supported by the DARPA GARD and DARPA QED4RML programs. Dickerson and Curry were supported in part by NSF CAREER Award IIS-1846237, DARPA GARD, DARPA SI3-CMD #S4761, DoD WHS Award #HQ003420F0035, and a Google Faculty Research Award. Additional support was provided by the National Science Foundation DMS division, and the JP Morgan Fellowship program.

## Footnotes

[1]When sorting by centers, we first sort vertically then horizontally. We found that this is important in achieving better results as in the object detection task, the horizontal location of a detected object seems to be more informative than the vertical location.

[2]For location binning, we split the image into 3x3 grid cells, and gather the corresponding boxes into bins based on the center of the box.

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
