[Supplementary Material]

# A Weaknesses of Mean Smoothing

In contrast to classification networks, where each output head encodes a function $f : \mathbb{R} \to [0, 1]$, the output of regression networks may vary over a larger range of values, e.g., between some lower and upper bounds $l$ and $u$. If we directly apply the same techniques for certifying classification output, based on bounding the gap between the highest and second-highest class probabilities, the resulting bounds on regression can be rather loose. To better articulate this claim, we derived the following result following the work in [1], as we recall from the main paper.

**Corollary 1.** *[1] For any $f : \mathbb{R}^d \to [l, u]$, the map $\eta(x) = \sigma \cdot \Phi^{-1}(\frac{g(x)-l}{u-l})$ is 1-Lipschitz, implying*

$$l + (u - l) \cdot \Phi \left( \frac{\eta(x) - \|\delta\|_2}{\sigma} \right) \leq g(x + \delta) \leq l + (u - l) \cdot \Phi \left( \frac{\eta(x) + \|\delta\|_2}{\sigma} \right) \qquad (1)$$

In light of the statement above, we make the following remarks.

**Location dependence.** Since $\Phi^{-1}$ is flatter around $0.5$ and steeper as it gets closer to $0$ or $1$, the non-linear Lipschitz bound in Corollary 1 is tightest when $g(x)$ is closer to $u$ or $l$ and loosest when $g(x) = \frac{u-l}{2}$. In the context of bounding-box regression, the coordinate-wise bound for the bounding box would be tighter when the sides of the bounding box are closer to the edges of the image, but looser when the sides are closer to the middle of the image. This strong bias in the tightness of the bound depending on the location of the box weakens the resulting worst-case bounds and makes the system more vulnerable to attacks targeting the middle portion of the output range.

**Skewness.** The mean-smoothed certificate is less sensitive to the shape of the distribution of $f(x + G)$. Intuitively, we would hope that the bound should be tighter for a more concentrated distribution compared to one that is more uniform. For example, the distribution could be significantly concentrated around a certain value in the support, but the mean-smoothed certificate only uses the expectation $\mathbb{E}[f(x + G)]$, which can be skewed by long tails and outliers.

Figure 1: Smoothing two functions with $G \sim N(0, 0.5^2)$: discrete (top) and continuous (bottom).

**Blurring.** When the base function $f$ outputs discrete values, its median smoothing will also stay discrete, whereas mean smoothing would be continuous; see the first row of Figure 1. Moreover, when the base function is continuous, median smoothing tends to be more similar to the original function; see the second row of Figure 1 where mean-smoothed outputs are attracted to 1, e.g., overestimating $f(2)$ and underestimating $f(3)$ in contrast to median smoothing which is more concentrated in the neighborhood of each input.

 **B   Proof of Lemma 2 - Adversarial Bounds for Percentile Smoothing**

29  Recall the definition of percentile smoothing as follows. Given a base function $f : \mathbb{R}^d \to \mathbb{R}$, with
30  $G \sim N(0, \sigma^2 I)$, we defined the *percentile smoothing* of $f$ as

$$\underline{h}_p(x) = \sup\{y \in \mathbb{R} \mid \mathbb{P}[f(x+G) \leq y] \leq p\} \tag{2}$$

$$\overline{h}_p(x) = \inf\{y \in \mathbb{R} \mid \mathbb{P}[f(x+G) \leq y] \geq p\} \tag{3}$$

31  where we use $h_p$ for convenience when the distinction is irrelevant.

32  In this appendix, we derive a bound on the variation in the percentile-smoothed function $h_p$ when the
33  input is corrupted by an adversarial perturbation $\delta$ of bounded $\ell_2$-norm. We do so by proving the
34  following statement, which we recall from the main paper.

35  **Lemma 2.** *A percentile-smoothed function $h_p$ with adversarial perturbation $\delta$ can be bounded as*

$$\underline{h}_{\underline{p}}(x) \ \leq h_p(x+\delta) \ \leq \overline{h}_{\overline{p}}(x) \quad \forall \, \|\delta\|_2 < \epsilon, \tag{4}$$

36  *where $\underline{p} := \Phi\left(\Phi^{-1}(p) - \frac{\epsilon}{\sigma}\right)$ and $\overline{p} := \Phi\left(\Phi^{-1}(p) + \frac{\epsilon}{\sigma}\right)$, with $\Phi$ being the standard Gaussian CDF.*

37  *Proof.* Consider the event $f(x+G) \leq \overline{h}_{\overline{p}}(x)$, where $G \sim N(0, \sigma^2 I)$, and let $\mathbb{1}_{f(x+G) \leq \overline{h}_{\overline{p}}(x)}$ be
38  the corresponding indicator function. We can treat the expectation of the indicator as a function of
39  $x$, which we denote by $\mathcal{E}(x) = \mathbb{E}[\mathbb{1}_{f(x+G) \leq \overline{h}_{\overline{p}}(x)}]$, and we have that $\mathcal{E} : \mathbb{R}^d \to [0, 1]$. Hence, the
40  mapping $x \mapsto \sigma \cdot \Phi^{-1}(\mathcal{E}(x))$ is 1-Lipschitz by Corollary 1 [1]. Noting that $\mathcal{E}(x) = \mathbb{P}[f(x+G) \leq$
41  $\overline{h}_{\overline{p}}(x)]$, we also have that

$$x \mapsto \sigma \cdot \Phi^{-1}(\mathbb{P}[f(x+G) \leq \overline{h}_{\overline{p}}(x)])$$

42  is 1-Lipschitz. It follows that under the perturbation by $\delta$, we have

$$\sigma \cdot \Phi^{-1}(\mathbb{P}[f(x+\delta+G) \leq \overline{h}_{\overline{p}}(x)]) \geq \sigma \cdot \Phi^{-1}(\mathbb{P}[f(x+G) \leq \overline{h}_{\overline{p}}(x)]) - \|\delta\|_2.$$

43  Rearranging, we get that

$$
\begin{aligned}
\Phi^{-1}(\mathbb{P}[f(x+\delta+G) \leq \overline{h}_{\overline{p}}(x)]) &\geq \Phi^{-1}(\mathbb{P}[f(x+G) \leq \overline{h}_{\overline{p}}(x)]) - \frac{\|\delta\|_2}{\sigma} \\
&\geq \Phi^{-1}(\mathbb{P}[f(x+G) \leq \overline{h}_{\overline{p}}(x)]) - \frac{\epsilon}{\sigma} && (\|\delta\|_2 \leq \epsilon) \\
&= \Phi^{-1}(\overline{p}) - \frac{\epsilon}{\sigma} && (\text{By the definition of } \overline{h}_{\overline{p}}(x)) \\
&= \Phi^{-1}(p) && (\text{By the definition of } \overline{p})
\end{aligned}
$$

44  By the monotonicity of $\Phi$, we get that

$$\mathbb{P}[f(x+\delta+G) \leq \overline{h}_{\overline{p}}(x)] \geq p$$

45  Recalling that $\overline{h}_p(x+\delta) = \inf\{y \in \mathbb{R} \mid \mathbb{P}[f(x+\delta+G) \leq y] \geq p\}$, we get that

$$\overline{h}_p(x+\delta) \leq \overline{h}_{\overline{p}}(x). \tag{5}$$

46  Similarly, it can be shown that for all $\|\delta\|_2 < \epsilon$, we have

$$\underline{h}_{\underline{p}}(x) \leq \underline{h}_p(x+\delta). \tag{6}$$

47  Combining the two bounds, and recalling the convenience notation of $h_p$, the proof follows. $\quad\square$

## C    Certified Precision and Recall for Varying $\ell_2$-Norm Bounds

To examine how the performance of our certified detector degrades as the adversary becomes stronger, we consider perturbations for larger values of $\epsilon$. As in the experiments reported in the main paper, we use the first 500 images of the MS-COCO dataset for testing on a pretrained YOLOv3 detector with an objectness threshold of 0.8 and an IoU threshold of 0.4. As in the main paper, we used an IoU threshold $\tau = 0.5$ for certification.

Table 1 below shows the certified precision and recall for $\|\delta\|_2 \leq \epsilon$, for varying values of $\epsilon$ compared to the setting $\epsilon = 0.36$ we used in the main paper. For the purposes of this comparison, we used location sorting with location and label binning. Note that the non-certified clean precision and recall obtained for this setup are $89.30\%$ and $16.07\%$, respectively.

| $\epsilon$ | Certified Precision | Certified Recall |
|---|---|---|
| 0.10 | 63.13% | 13.46% |
| 0.25 | 41.39% | 10.79% |
| 0.36 | 28.86% | 9.10% |
| 0.50 | 15.82% | 6.78% |

Table 1: Certified precision and recall for different bounds on the perturbation $\|\delta\|_2 \leq \epsilon$.

Note that the certified precision drops much faster because the maximum number of possible predictions increases quite quickly as $\epsilon$ becomes larger.

 # D  Detailed Precision-Recall Curve for AP calculation

| Conf. Thresh. | Sorting | Binning | Denoise | Precision | Recall | Certified Precision | Certified Recall |
|---|---|---|---|---|---|---|---|
| 0.8 | Objectness | None | No | 50.27% | 8.01% | 8.08% | 2.30% |
| 0.6 | Objectness | None | No | 38.72% | 9.74% | 6.17% | 2.60% |
| 0.4 | Objectness | None | No | 28.75% | 10.76% | 4.50% | 2.69% |
| 0.2 | Objectness | None | No | 18.24% | 11.68% | 2.79% | 2.70% |
| 0.1 | Objectness | None | No | 11.28% | 11.94% | 1.74% | 2.71% |
| 0.8 | Location | None | No | 53.81% | 8.57% | 7.70% | 2.19% |
| 0.6 | Location | None | No | 42.28% | 10.64% | 4.92% | 2.07% |
| 0.4 | Location | None | No | 31.51% | 11.79% | 2.73% | 1.63% |
| 0.2 | Location | None | No | 18.97% | 12.15% | 1.09% | 1.06% |
| 0.1 | Location | None | No | 10.87% | 11.51% | 0.42% | 0.66% |
| 0.8 | Objectness | Label | No | 58.40% | 8.44% | 8.07% | 2.85% |
| 0.6 | Objectness | Label | No | 47.52% | 10.62% | 6.29% | 3.48% |
| 0.4 | Objectness | Label | No | 37.41% | 12.25% | 4.71% | 3.88% |
| 0.2 | Objectness | Label | No | 25.65% | 14.25% | 2.98% | 4.17% |
| 0.1 | Objectness | Label | No | 17.04% | 15.70% | 1.93% | 4.42% |
| 0.8 | Location | Label | No | 61.96% | 8.94% | 9.27% | 3.26% |
| 0.6 | Location | Label | No | 51.42% | 11.47% | 6.88% | 3.80% |
| 0.4 | Location | Label | No | 41.13% | 13.45% | 4.56% | 3.76% |
| 0.2 | Location | Label | No | 28.41% | 15.80% | 2.43% | 3.41% |
| 0.1 | Location | Label | No | 18.63% | 17.18% | 1.26% | 2.88% |
| 0.8 | Objectness | Location | No | 58.25% | 8.76% | 10.01% | 3.27% |
| 0.6 | Objectness | Location | No | 47.74% | 11.21% | 7.72% | 3.95% |
| 0.4 | Objectness | Location | No | 38.35% | 13.21% | 5.75% | 4.35% |
| 0.2 | Objectness | Location | No | 26.09% | 15.44% | 3.68% | 4.64% |
| 0.1 | Objectness | Location | No | 17.18% | 17.03% | 2.40% | 4.85% |
| 0.8 | Location | Location | No | 59.44% | 8.90% | 9.88% | 3.23% |
| 0.6 | Location | Location | No | 48.24% | 11.34% | 7.01% | 3.58% |
| 0.4 | Location | Location | No | 38.87% | 13.39% | 4.59% | 3.47% |
| 0.2 | Location | Location | No | 26.20% | 15.51% | 2.25% | 2.84% |
| 0.1 | Location | Location | No | 16.90% | 16.73% | 1.08% | 2.18% |
| 0.8 | Objectness | Location+Label | No | 63.48% | 8.79% | 9.26% | 3.52% |
| 0.6 | Objectness | Location+Label | No | 52.92% | 11.21% | 7.17% | 4.37% |
| 0.4 | Objectness | Location+Label | No | 43.96% | 13.38% | 5.37% | 5.00% |
| 0.2 | Objectness | Location+Label | No | 31.62% | 15.92% | 3.42% | 5.56% |
| 0.1 | Objectness | Location+Label | No | 21.79% | 18.07% | 2.17% | 5.90% |
| 0.8 | Location | Location+Label | No | 64.45% | 8.91% | 9.78% | 3.72% |
| 0.6 | Location | Location+Label | No | 53.70% | 11.40% | 7.45% | 4.54% |
| 0.4 | Location | Location+Label | No | 44.99% | 13.68% | 5.32% | 4.95% |
| 0.2 | Location | Location+Label | No | 32.63% | 16.46% | 2.95% | 4.80% |
| 0.1 | Location | Location+Label | No | 22.41% | 18.59% | 1.63% | 4.43% |

| Conf. Thresh. | Sorting | Binning | Denoise | Precision | Recall | Certified Precision | Certified Recall |
|---:|---|---|---|---:|---:|---:|---:|
| 0.8 | Objectness | None | Yes | 75.12% | 17.69% | 18.51% | 6.00% |
| 0.6 | Objectness | None | Yes | 67.06% | 20.38% | 15.44% | 6.45% |
| 0.4 | Objectness | None | Yes | 58.76% | 22.41% | 12.71% | 6.74% |
| 0.2 | Objectness | None | Yes | 46.70% | 24.49% | 9.50% | 6.95% |
| 0.1 | Objectness | None | Yes | 35.69% | 25.45% | 7.05% | 7.03% |
| 0.8 | Location | None | Yes | 83.85% | 19.75% | 21.01% | 6.80% |
| 0.6 | Location | None | Yes | 76.12% | 23.12% | 16.55% | 6.92% |
| 0.4 | Location | None | Yes | 66.59% | 25.41% | 11.91% | 6.30% |
| 0.2 | Location | None | Yes | 51.92% | 27.23% | 7.09% | 5.19% |
| 0.1 | Location | None | Yes | 37.82% | 27.00% | 3.82% | 3.80% |
| 0.8 | Objectness | Label | Yes | 84.51% | 19.47% | 24.08% | 8.57% |
| 0.6 | Objectness | Label | Yes | 78.25% | 23.01% | 20.86% | 9.89% |
| 0.4 | Objectness | Label | Yes | 71.65% | 26.19% | 17.43% | 10.77% |
| 0.2 | Objectness | Label | Yes | 60.98% | 29.96% | 12.94% | 11.68% |
| 0.1 | Objectness | Label | Yes | 50.17% | 32.89% | 9.52% | 12.34% |
| 0.8 | Location | Label | Yes | 90.04% | 20.75% | 28.72% | 10.24% |
| 0.6 | Location | Label | Yes | 84.78% | 24.96% | 24.92% | 11.81% |
| 0.4 | Location | Label | Yes | 78.38% | 28.63% | 20.04% | 12.38% |
| 0.2 | Location | Label | Yes | 67.37% | 33.10% | 13.49% | 12.17% |
| 0.1 | Location | Label | Yes | 54.96% | 36.03% | 8.52% | 11.04% |
| 0.8 | Objectness | Location | Yes | 87.24% | 20.13% | 26.65% | 9.63% |
| 0.6 | Objectness | Location | Yes | 81.75% | 24.26% | 23.12% | 10.98% |
| 0.4 | Objectness | Location | Yes | 74.65% | 27.61% | 19.36% | 11.85% |
| 0.2 | Objectness | Location | Yes | 63.21% | 31.73% | 14.58% | 12.67% |
| 0.1 | Objectness | Location | Yes | 50.83% | 34.57% | 10.89% | 13.21% |
| 0.8 | Location | Location | Yes | 89.67% | 20.68% | 26.83% | 9.70% |
| 0.6 | Location | Location | Yes | 84.06% | 24.95% | 22.53% | 10.71% |
| 0.4 | Location | Location | Yes | 77.28% | 28.56% | 17.66% | 10.81% |
| 0.2 | Location | Location | Yes | 65.19% | 32.74% | 11.67% | 10.14% |
| 0.1 | Location | Location | Yes | 51.71% | 35.17% | 7.26% | 8.81% |
| 0.8 | Objectness | Location+Label | Yes | 90.42% | 20.54% | 27.73% | 10.69% |
| 0.6 | Objectness | Location+Label | Yes | 85.47% | 24.74% | 24.28% | 12.49% |
| 0.4 | Objectness | Location+Label | Yes | 79.61% | 28.47% | 20.41% | 13.76% |
| 0.2 | Objectness | Location+Label | Yes | 69.52% | 33.05% | 15.15% | 15.04% |
| 0.1 | Objectness | Location+Label | Yes | 58.13% | 36.49% | 11.06% | 16.04% |
| 0.8 | Location | Location+Label | Yes | 91.93% | 20.87% | 29.73% | 11.44% |
| 0.6 | Location | Location+Label | Yes | 87.33% | 25.30% | 25.87% | 13.32% |
| 0.4 | Location | Location+Label | Yes | 81.72% | 29.23% | 21.29% | 14.36% |
| 0.2 | Location | Location+Label | Yes | 71.90% | 34.22% | 15.11% | 15.02% |
| 0.1 | Location | Location+Label | Yes | 60.31% | 37.85% | 10.14% | 14.71% |

## References

[1] Hadi Salman, Jerry Li, Ilya Razenshteyn, Pengchuan Zhang, Huan Zhang, Sebastien Bubeck, and Greg Yang. Provably Robust Deep Learning via Adversarially Trained Smoothed Classifiers. In *Advances in Neural Information Processing Systems*, 2019.