[Reviews · NeurIPS 2020]

Review 1

Summary and Contributions: This paper considers the problem of designing a certifiably robust object detector. One issue with applying existing randomized smoothing techniques is that object detection is essentially a regression problem, and Gaussian mean smoothing gives very loose bounds when the base function has outputs that can vary over a wide range. To remedy this issue, this paper proposes "median smoothing," in which the prediction of the smoothed function at some point x is defined as the median (rather than the mean) of the random variable f(x+noise). The key result is that for any perturbed point within L2 distance \epsilon of x, the median-smoothed function is lower- and upper-bounded by the p1 and p2-th percentiles of the random variable f(x+noise), where p1 and p2 are numbers in [0, 1] that depend on \epsilon and the noise variance. The paper shows how to apply this technique to object detection.

Strengths: The median smoothing technique is interesting and potentially applicable to regression problems in many other domains.

Weaknesses: Much of the paper (Section 5) is about various tricks needed to get the proposed method to work for the specific application of object detection. This part feels much better suited to a vision conference than to ICLR.

Correctness: The claims are reasonable and the method seems likely to be correct.

Clarity: Yes.

Relation to Prior Work: Yes.

Reproducibility: Yes

Additional Feedback: Update: I've read the rebuttal, and I will keep my score.


Review 2

Summary and Contributions: This paper proposes a method to probabilistically defend and certify object detectors against l2 adversaries. This is achieved by reducing object detection to a regression task and by introducing median smoothing which is less vulnerable to outliers compared to mean smoothing. Further, the proposed methods are evaluated.

Strengths: The authors propose a certification and defense algorithm against l2 adversaries, suited for object detection. Certifying the robustness of object detectors and defending them is an important problem, as object detection can have many (safety critical) applications.

Weaknesses: - Equation 3: To me it looks as if equation 3 is not well defined. For the function f: R^d -> R sending all x \in R^d to 0, we have that f(x + G) = 0 always holds. Thus max_y y s.t. P(f(x+G) <= y) <= p is equivalent to max_y y s.t. [0 <= y] <= p. If y >= 0, then the term [0 <= y] <= p is satisfied only if p = 1. Further, for any y < 0, term is not satisfied. Hence for any p \in [0,1[, max_y y s.t. P(f(x+G) <= y) <= p is equivalent to max_y y s.t. y < 0, which is in standard analysis not well defined. That said, i think the authors have multiple options to fix this. - The certification rates for the respective radius are too low for practical use cases as the authors acknowledge.

Correctness: The claims seem to hold and the method seems to work. The empirical methodology seems to be correct.

Clarity: Overall the paper is mostly well written and easy to read.

Relation to Prior Work: Line 56-57: Here, the authors seem to confuse defending with certifying. Some works doing certification are - Katz et.al: "Reluplex: An efficient SMT solver for verifying deep neural networks" - Raghunathan et.al: "Semidefinite relaxations for certifying robustness to adversarial examples" - Gehr et.al: "AI2: Safety and Robustness Certification of Neural Networks with Abstract Interpretation" - Zhang et.al: "Efficient neural network robustness certification with general activation functions" - Singh et.al: "An abstract domain for certifying neural networks"

Reproducibility: Yes

Additional Feedback: - Line 90-97: The term "worst case bounding box" here should be defined clearly and early. - Line 131: Do the authors mean here “are predicted” or “can be predicted”? - Figure 2: It would improve the figure if the figure would explicitly show r, b and d. - The authors should here provide the rate of successful attacks for a powerful object detection adversary to put the results of section 6 into more context. Ideally, the authors design an attack tailored to their defense (similar to [24]). ################################# Update after reading the rebuttal: I raised my score.


Review 3

Summary and Contributions: This paper is the first work that studies defenses against adversarial attacks, namely l2-bounded, for object detectors. Since object detector is more complicate and the prediction of bounding-box is a regression task, standard defense methods for image classification task are difficult to be applied directly. This paper treats the whole detection pipeline as a black-box without requiring re-training. Due to the large range of output values in regression problems, previous certificates for classification problem based on mean values become weaker. To the end, this paper presents a novel certificate based on the median of smoothed predictions which is less susceptible to outliers. Meanwhile, it also provides the theoretical analysis of weaknesses of mean smoothing and adversarial bounds for percentile smoothing. Experiments on YOLOv3 object detector could justify the effectiveness of the proposed method to some degree.

Strengths: —The paper is well written; the details of the method are described clearly. —It is the first work that focuses on the defenses against adversarial attacks by interpreting the object detection as a regression problem. An effective yet simple median smoothing is presented for the problem. —Theoretical proofs are convincing to justify the weakness of the mean smoothing and motivations of the proposed method.

Weaknesses: More experiments are expected. The experiment part only works on a single object detector while at least two detectors with different frameworks are expected such as the typical two-stage detector Faster-RCNN. Additionally, the whole evaluation is done on 1/10 images from validation set of MS-COCO dataset. Why do not use all of them?

Correctness: YES

Clarity: YES

Relation to Prior Work: YES

Reproducibility: Yes

Additional Feedback: Update: I have read all the reviews and rebuttal, and the authors have answered my questions well.


Review 4

Summary and Contributions: The authors use an algorithm based around ideas from randomized smoothing to certify lp-robustness in object detection. The authors develop methods on top of randomized smoothing to deal with (a) continuous outputs and (b) variable length outputs. The work presents (as far as I know) the first method for certifiably lp-robust object detection. I would vote for acceptance as long as the major issues pointed out below are resolved.

Strengths: This paper presents the first certified method for lp-robustness on object detection algorithms. The core mechanic in the algorithm, taking the median of predictions from randomly perturbed inputs, relies on a basic concentration argument. The variable length output of object detection methods is dealt with by sorting, binning, and padding sequences of output boxes. Finally, the method works with arbitrary object detectors as it just requires black-box input/output access.

Weaknesses: The paper has some issues in evaluation --- the authors estimate held-out performance in a way that could not be representative of the true held-out performance, draw comparisons between algorithm performances with differences in magnitude that are smaller than the noise induced by the performance estimation method, do not use standard object detection metrics, and do not discuss inference speed at all. The paper does not properly compare with previous work, and is confusingly written at times. All points here are expounded on in the sections below.

Correctness: Evaluation completeness: The authors only evaluate on the first 500/5000 of test set images. This is not necessarily a random sampling (depending on how the test set images are ordered), and the authors should either randomly evaluate on a subset of test set images or (ideally) the full test set to accurately gauge task performance. Confidence intervals: because the authors only evaluate on 500/5000 images, even in the best case assuming that these are randomly sampled the confidence intervals for (for example) recall would be larger than the differences measured between all the different certified sorting methods---the largest difference is only ~1.2%. The full test set must be evaluated on for these metrics to be meaningful. Evaluation of object detection methods: in object detection there are standard metrics like AP@[IOU threshold] that capture precision/recall at various thresholds for "objectness." It would be good to include metrics (both for the certified and non-certified methods) that account for varying the IOU threshold, like mAP@50, mAP@30, mAP@70 as these are the standard metrics by which object detection methods are judged (for better or for worse). It would also be good to include standard object detection methods as a baseline here (e.g. recreate something like Figure 1 in [1], with the inference time axis as well). Discussion on inference speed: it would be good to discuss inference speed. The described method would be much slower due to the voting procedure, and inference speed is considered crucial in object detection (due to applications like self-driving cars that have a maximum processing speed requirement). How many perturbed images are used to compute a given certificate? [1] https://pjreddie.com/media/files/papers/YOLOv3.pdf

Clarity: The paper is sometimes not written very well. The motivation for the "Encoding detectors as regressors." section is unclear and I had to read it a few times to get the full picture. There is no central figure with inference speed vs task AP for this method vs other object detection methods, only Table 1 to 3. The tables are sometimes unclear; just in Table 1, the first row's first column overflows into the second column and uses a generic name "Blackbox Detector" instead of the actual detector used (Yolov3). Generally series of numbers are hard to interpret and these tables could be better formatted graphically. Lines 16-17 either has poor grammar or claims that object detectors are more "realistic" than classification systems; these are different settings and one is not more inherently realistic than the other.

Relation to Prior Work: Related work: this paper claims to present the first adversarial defense for object detectors, but there has been previous work in this area (e.g. [0], ICCV 2019). It would be good for the authors to do a comprehensive literature search and provide comparisons to the presented method (and also to amend the wording in the paper). [0] https://openaccess.thecvf.com/content_ICCV_2019/papers/Zhang_Towards_Adversarially_Robust_Object_Detection_ICCV_2019_paper.pdf

Reproducibility: Yes

Additional Feedback: Varying object detectors: it would be interesting to see how the resulting robustness changes as the performance of the base object detector changes. ----------------- Post response feedback: Thank you for addressing my concerns on related work. I would vote for acceptance, raising my score to a 6, if in the final version the authors (a) evaluate on their entire test set (updating numbers appropriately, including comparisons between different sorting methods), (b) add the discussion on inference speed, and (c) add the AP metrics.

[Author Response · NeurIPS 2020]

**Evaluation.**  Our initial experiments only used the first 500 images from the MS-COCO data set, as was done in [3]. In the tables below we use the full test set with the best performing reduction based on the initial experiments. As **R3** and **R4**  suggest, we will use the full test set to update the comparison Table 1 of Sec. 6 toawrds the final manuscript.

**Additional Experiments.**  As **R3** suggested, we provide additional results for different architectures. While the median-smoothed RCNN has better clean performance as measured by F1 score, its certified precision recall is strictly lower highlighting the trade-off between clean accuracy and robustness as discussed in the literature.

|  | Precision | Recall | F1 | Certified Precision | Certified Recall |
|---|---|---|---|---|---|
| **YOLOv3** | 91.80% | 20.88% | 34.02% | 29.65% | 11.43% |
| **Faster RCNN** | 86.58% | 24.63% | 38.35% | 17.69% | 10.85% |
| **Mask RCNN** | 85.50% | 25.14% | 38.85% | 17.42% | 10.85% |

**Other Performance Metrics.**  As **R4** suggested, we examine the average precision (AP) for both the plain and certified detectors; we report the results for YOLOv3 below, and plan to include Faster RCNN and Mask RCNN in the final manuscript. Since varying the objectness threshold changes the base detector ($f$), we reevaluate the smoothed detector at objectness thresholds $\{0.1, 0.2, 0.4, 0.6, 0.8\}$ and calculate the area under the steps to lower bound true AP. As for inference speed, the smoothing paradigm is inherently costly, as we use 2000 perturbations for Monte Carlo estimation. We leave it to future work to improve these important metrics as needed in practice.

|  | AP@50 | Certified AP@50 |
|---|---|---|
| YOLOv3 | 32.0% | 4.2% |

**Comparison to Prior Work.**  We thank **R4**  for bringing relevant prior work to our attention. While we cannot claim to be the first adversarial defense for object detection, we maintain that we provide the first certified defense. The performance of the certified defense approach is currently so weak compared to the adversarial training approach that we do not think a meaningful quantitative comparison can be done. We will make sure to discuss the relation to this prior work as follow: "our certified radius is 0.36 in terms of $\ell_2$-norm whereas Zhang et al. (ICCV'19) achieved robustness radius of 8/255 in terms of the stronger $\ell_\infty$-norm threat model." As **R2** suggests, we will distinguish certification and defense as follows: "Several methods of obtaining robustness certificates for classification problems have been proposed [19, 21, XX]. In addition, [9, 11, 24, 25, 29, YY] proposes methods to both defend the model while enabling better certificates;" we will make sure to include the citations suggested by **R2**.

**Tailored Empirical Attacks.**  As **R2** suggested, we implemented a DAG attack against our best performing model. The DAG attack is modified to include Monte Carlo sampling to increase the strength of the attack. We take 20 PGD steps and draw 5 random samples to estimate the gradient of the smoothed model. Surprisingly, the smoothed model is quite robust within the desired radius. The DAG attack was only able to decrease recall by 1.1%. This illustrates that the bound we obtained is likely quite loose with respect to the true robustness of the object detector, and we leave improvements of the robustness certificate as future work.

**Certifiable radius.**  **R2** rightly points out that as of right now the certifiable radius is too low for real-world applications. We emphasize that certified robustness is a challenging domain and none of the existing methods yield practical certificates even for classification problems. For example, while the SOTA certified defense by Salman et al. (NeurIPS'19) achieved 68.2% certified accuracy for $||\epsilon||_\infty < 2/255$ on CIFAR-10, the empirical approach of Xie et al. (CVPR'19) can achieve similar empirical robustness but at the ImageNet scale and with larger radius. That said, our work leverages a principled smoothing approach to provide the first non-trivial certificates for architectures as complex as object detectors.

**Generality of the Techniques.**  **R1** remarks that the sorting and bucketing techniques proposed in Section 5 may be too specific for object detection. We note that the proposed techniques are potentially applicable to certifying other networks through reductions to a regression formulation. This is particularly relevant for tasks that have variable length outputs, such as key points detection, instance segmentation, or image captioning. Viewed in the broader context of adversarial robustness for ML models, computer vision continues to provide exemplar problems and we hope our work on object detection will help advance both the theory and practice of this important field.

**Other Clarifications and Corrections.**  **R2** correctly points out an issue with Equation 3. We already fixed this issue in the supplemental materials, replacing min/max with inf/sup. We will also clarify the definition of the worst-case bounding box as "the box with coordinates satisfying the certified upper and lower bounds which realizes the lowest

[Meta-Review · NeurIPS 2020]

The work proposes a simple and effective approach to certify object detection as a regression problem. The theoretical proofs that show the weakness of mean smoothing and effectiveness of the method are interesting. The approach also has potential to be used in other domains and applications. The author response was critical in the decision to accept this paper and next time I would recommend the authors evaluate on the full test set before paper submission. Overall the work would be a good contribution to the conference assuming the full test set evaluation, AP metrics and a discussion on inference speed are included.